# Revision of Cytomegalovirus Immunoglobulin M Antibody Titer Cutoff in a Maternal Antibody Screening Program in Japan: A Cohort Comparison Involving a Total of 32,000 Pregnant Women

**DOI:** 10.3390/v15040962

**Published:** 2023-04-13

**Authors:** Asa Kitamura, Kuniaki Toriyabe, Miki Hagimoto-Akasaka, Kyoko Hamasaki-Shimada, Makoto Ikejiri, Toshio Minematsu, Shigeru Suga, Eiji Kondo, Masamichi Kihira, Fumihiro Morikawa, Tomoaki Ikeda

**Affiliations:** 1Department of Obstetrics and Gynecology, Mie University Graduate School of Medicine, Tsu 514-8507, Japan; 2Department of Obstetrics and Gynecology, National Hospital Organization Mie Chuo Medical Center, Tsu 514-8507, Japan; 3Department of Obstetrics and Gynecology, Kurashiki Medical Center, Okayama 710-8522, Japan; 4Clinical Laboratory, Mie University Hospital, Tsu 514-8507, Japan; 5Research Center for Disease Control, Aisenkai Nichinan Hospital, Miyazaki 887-0034, Japan; 6Institute for Clinical Research, National Hospital Organization Mie National Hospital, Tsu 514-8507, Japan; 7Mie Association of Obstetricians and Gynecologists, Tsu 514-8507, Japan

**Keywords:** cytomegalovirus, IgM antibody, serological screening, maternal antibody screening, primary infection, congenital infection

## Abstract

Cytomegalovirus (CMV) is associated with congenital infections. We aimed to validate the revised CMV immunoglobulin (Ig) M titer cutoff for IgG avidity measurements as a reflex test in maternal screening to identify women with primary CMV infection and newborn congenital cytomegalovirus (cCMV). We screened maternal CMV antibodies (the Denka assay) in Japan, from 2017 to 2019, using a revised IgM cutoff (≥4.00 index). Participants were tested for IgG and IgM antibodies, and for IgG avidity if IgM levels exceeded the cutoff. We compared these with corresponding results from 2013 to 2017 based on the original cutoff (≥1.21) and recalculated using the revised cutoff. Newborn urine CMV DNA tests were performed for women with low avidity (≤35.0%). Among 12,832 women screened in 2017–2019, 127 (1.0%) had IgM above the revised cutoff. Thirty-five exhibited low avidity, and seven infants developed cCMV. Of 19,435 women screened in 2013–2017, 184 (1.0%) had IgM above the revised cutoff, 67 had low avidity, and 1 had cCMV. The 2017–2019 results were not significantly different from the 2013–2017 results. The revised IgM cutoff improves maternal screening in identifying primary infection and newborn cCMV; however, further study related to other assays than Denka is required.

## 1. Introduction

Maternal infections during pregnancy that can cause congenital abnormalities in infants include those from the TORCH complex: *Toxoplasma gondii*, rubella virus, cytomegalovirus (CMV), herpes simplex virus, and others (e.g., *Treponema pallidum* and parvovirus B19). CMV (Order *Herpesvirales*, Family *Herpesviridae*, Subfamily *Betaherpesvirinae*, Genus *Cytomegalovirus*) is the most common pathogen associated with mother-to-child transmission through the placenta and causes damage to the fetal nervous system. On average, 2.7 million new cases of congenital CMV (cCMV) infection occur worldwide each year [1]. The living environment of pregnant women influences incidence: cCMV is associated with maternal exposure in the household through nursery school children, sharing of feeding cups, and sexually transmitted infections [2].

Reactivation, reinfection, and primary CMV infection during pregnancy can cause infant cCMV, which can lead to neurological sequelae, the most frequent being sensorineural hearing loss (SNHL), intellectual disability, cerebral palsy, visual defects, psychomotor impairments, and seizures. cCMV infection is the most frequent viral cause of intellectual disability and leading non-genetic cause of congenital SNHL in infancy. Congenital SNHL caused by cCMV may become apparent in older children who were not diagnosed in universal newborn hearing screenings and, therefore, did not undergo neonatal anti-viral drug treatment. Infants with cCMV should, thus, be followed up until 6 years of age for earlier intervention and better management and to control long-term neurological sensorineural sequelae [3].

Serological screenings, including assessments of CMV immunoglobulin (Ig) G and IgM antibody levels, are used to identify recent primary infections. Maternal CMV antibody screenings are mainly performed to identify primary infections, which are associated with a higher risk of cCMV transmission than non-primary infections. The risk of occurrence of symptoms related to cCMV infection is highest after maternal primary infection in the first trimester of pregnancy. CMV IgG avidity tests are also used in maternal antibody screenings [4]. Seroconversion to IgG antibody positivity proves maternal primary infection, but the combination of positive IgG and IgM antibodies and low IgG avidity is not always proof [5].

Usually, only the presence or absence of IgM antibodies is considered in screenings. However, even when used in combination with IgG avidity, IgM tests are insufficient for the effective diagnosis of primary infections. Employing the Denka CMV IgM assay (Denka Co., Ltd., Tokyo, Japan), which is widely used in Japan, we previously reported the usefulness of IgM assay titers in predicting low maternal IgG avidity and the occurrence of infant cCMV [6]; others have reported the potential usefulness of the same assay [7]. These reports suggested a revised CMV IgM titer cutoff in maternal CMV antibody screenings in Japan, which might be used to test CMV IgG avidity as a reflex test to detect maternal primary CMV infection and infant cCMV infection.

Despite the possible usefulness of CMV IgM titers in predicting low maternal IgG avidity and the occurrence of infant cCMV, the validity of the revised CMV IgM titer cutoff compared with that of the original cutoff in maternal serological screenings has not been examined. Here, we prospectively examined the validity of the revised cutoff relative to the original cutoff of the CMV IgM antibody titer in a maternal serological screening to identify pregnant women with primary CMV infections and subsequent cCMV.

## 2. Materials and Methods

### 2.1. Antibody and DNA Tests

Serum obtained during pregnancy (early-stage, ≤20, and late-stage, ≥28 gestational weeks) was used for antibody tests. Negative, equivocal, and positive results were defined as IgG titers (kit-specific CMV IgG antibody value) of ≤1.9, 2.0–3.9, and ≥4.0, respectively, and IgM titers (kit-specific CMV IgM antibody index) of ≤0.79, 0.80–1.20, and ≥1.21 (the original cutoff for IgG avidity measurements as a reflex test) in Denka CMV assays, which are enzyme-linked immunosorbent assays [8]. IgG antibody avidity was measured at Aisenkai Nichinan Hospital (Nichinan, Japan), using the Enzygnost CMV IgG assay (Siemens Healthcare Diagnostics K.K., Tokyo, Japan) as previously reported [6,9]. An avidity index of ≤35.0% was considered low, and participants with low avidity were considered as having a primary infection during early pregnancy or high risk of subsequent cCMV. An avidity index of ≥35.1% was considered high, and participants with high avidity were considered as having a non-primary infection or low risk of cCMV. Avidity of IgG antibody was measured using IgG assays rather than assays specifically designed for IgG avidity. Therefore, these low or high avidity indexes were determined by laboratory criteria [6].

Neonatal urine obtained during the first week after birth was used for CMV DNA testing (gene symbol, UL123; gene description, regulatory protein IE1), and real-time polymerase chain reaction (PCR) was performed at Mie University Hospital (Tsu, Japan) [6,9]. Real-time PCR was performed using an ABI 7900 Fast Real-time PCR system (Applied Biosystems, Foster City, CA, USA) in a total volume of 25 μL, consisting of 12.5 μL of QuantiTect Multiplex PCR Kit (Qiagen, Hilden, Germany), 1.0 μL each of 10 μM sense (5′-TTTTTAGCACGGGCCTTAGC-3′) and antisense (5′-AAGGAGCTGCATGATGTGACC-3′) primers, 2.56 μM TaqMan Probe (5′-FAM-TGCAGTGCACCCCCCAACTTGTT-TAMRA-3′) (Applied Biosystems), and 10 μL of the sample DNA. Viral DNA was extracted from samples using a QIAamp DNA Blood Mini Kit (Qiagen). DNA was amplified with a precycling hold at 50 °C for 2 min, followed by 45 cycles at 95 °C for 15 s and 58 °C for 60 s. As the annealing and extension steps were performed at the same temperature (58 °C), these steps were simultaneously performed for a total of 60 s. Samples were analyzed in duplicate. The AcroMetrix CMVtc Panel (Applied Biosystems) was used to generate the standard curve. Infants with ≥200 copies/mL CMV DNA in fresh urine were diagnosed with cCMV. In cases of infection, additional viral isolation and subunit analysis of glycoprotein B (gB) were conducted using the same sample at Mie National Hospital (Tsu, Japan) [9,10]. In subunit analysis of gB, another real-time PCR was performed. A mixture of both gB134 (5′-TTCTGGCAAGGTATCAAGCA-3′) and gB2 (5′-TGTTCTGGCAAGGCATCA-3′) forward primers and both gB124 (5′-AACTGCAGCTGGGCGTA-3′) and gB3 (5′-TGAACTGCAGCTGAGCGTA-3′) reverse primers were used. A reaction mixture measuring 20 μL contained 10 μL of 2 X TaqMan Fast Advanced Master Mix (Applied Biosystems), 0.4 μL of each 10 μM primer, 0.4 μL of 10 μM probe for gB1-4, 2 μL of the sample DNA, and 6 μL of diluted water in separate wells. DNA amplification was performed at 95 °C for 20 s, followed by 40 cycles at 95 °C for 1 s and at 60 °C for 20 s using an ABI StepOne Plus real-time PCR system (Applied Biosystems). The threshold cycle value for each sample was determined, and the genotype of the sample was identified using StepOne Software v2.3.

### 2.2. Maternal Screening Program and Neurological Tests in cCMV Infants

The maternal antibody screening program “Cytomegalovirus in Mother and infant-engaged Virus serology (CMieV)” was launched in Mie, Japan, in 2013. This study was approved by the Clinical Research Ethics Review Committee of Mie University Hospital (approval number: 2610), and all participants provided written informed consent. This prospective cohort study was conducted in accordance with the Declaration of Helsinki. Early-stage pregnant women were registered at obstetric institutions and tested for serum CMV IgG and IgM antibodies at ≤20 gestational weeks. From September 2013 to March 2017, we used a cutoff of ≥1.21 IgM antibody index (the original cutoff) to measure IgG avidity as a reflex test, and from April 2017 to March 2019, we used a cutoff of ≥4.00 index (the revised cutoff). Although prior studies suggested possible revised IgM cutoffs of 6.00 or 7.28 index [6,7], to eliminate the risk of missing the takeout, this study adopted the cutoff of ≥4.00 index, one value lower than for the reported IgM titer.

Pregnant women were divided into three groups according to their antibody levels. (1) In women with positive/equivocal IgG and IgM above the cutoffs (original or revised cutoff during the corresponding periods), IgG avidity was measured as a reflex test using the same serum sample. Women exhibiting low IgG avidity were considered as having a primary infection; neonatal urine CMV DNA tests were performed after delivery (Figure 1). (2) In women exhibiting high IgG avidity or IgM under the cutoff results (the former period, <1.21, and the latter period, <4.00 of IgM titer index), no further antibody nor neonatal urine tests were performed with the exception of some. (3) During both periods, in women with negative IgG results, precautionary measures, such as avoiding close contact with the urine or saliva of young children and usage of condoms during sexual intercourse, were taken to avoid primary infections. Repeated IgG and IgM tests were performed at ≥28 gestational weeks. Women exhibiting IgG antibody seroconversion (negative to positive) were considered as having a primary infection; neonatal urine CMV DNA tests were performed after delivery.

In infants with cCMV infections, neurological tests, such as brain magnetic resonance imaging (MRI), auditory brainstem response (ABR) tests, and funduscopic examinations, were performed approximately 1 month after birth. cCMV infants with abnormalities in these neurological tests were diagnosed with symptomatic cCMV and underwent anti-CMV therapy, whereas those without abnormalities were diagnosed with asymptomatic cCMV and were not treated.

### 2.3. Comparison of Maternal Screening with the Original and Revised Cutoffs of CMV IgM Antibody Titer

We retrospectively re-analyzed laboratory data obtained based on the original CMV IgM cutoff, using the revised cutoff, as below. First, to confirm the validity of the revised cutoff of CMV IgM antibody titer as part of the screening program, we assessed the difference in the proportions of women with positive/equivocal IgG and IgM above cutoffs (original and revised) among all women. Second, we assessed the differences in the proportions of women with IgM above the revised cutoff, low IgG avidity, congenital infections including symptomatic infection, or symptomatic congenital infections among women with IgM above the original cutoff results. Third, to confirm the effectiveness of the revised cutoff, we studied sensitivity, specificity, positive predictive value, negative predictive value, and accuracy for cCMV. Moreover, we compared the numbers of IgG antibody avidity tests required to detect one congenital infection case including symptomatic infection or one symptomatic congenital infection case.

Differences were analyzed using Fisher’s exact test. *p*-values < 0.05 were considered statistically significant. All analyses were conducted using IBM SPSS Statistics 27 (IBM Corp., Armonk, NY, USA).

## 3. Results

### 3.1. Maternal Screening with Revised Cutoff of CMV IgM Antibody Titer and Infant cCMV

In total, 12,832 pregnant women were registered at 24 institutions from April 2017 to March 2019 (the revised cutoff period). Median values of age, parity, and gestational week at testing were 30 (range 16–51) years, 1 (0–7), and 11 (4–20) weeks. The result after April 2017 is shown in Appendix A and the sum results of before and after April 2017 are shown in Figure 2 and Appendix A, respectively.

Among the participants, 127 (1.0%) women had positive/equivocal IgG and IgM above the revised cutoff results. Thirty-five of these exhibited low avidity, and seven infants, who were all live births, had congenital infections. Six of the seven were positive for both CMV DNA and viral isolation in neonatal urine, whereas one was only positive for CMV DNA. Two cases were symptomatic. One showed a cerebral calcification on brain MRI and a unilateral threshold elevation on ABR testing, whereas the other showed a bilateral threshold elevation. Both underwent drug therapy (Table 1). Among 12,832 women, 7795 (60.7%) had positive/equivocal IgG and negative/equivocal IgM results or IgM under the revised cutoff results. The numbers of women, neonatal urine CMV DNA tests, and cCMV cases before and after April 2017 according to the results of maternal screening other than low avidity are shown in Table 2.

Among all 12,832 women, 4910 (38.1%) had negative IgG antibody results. Among these, 2941 underwent IgG and IgM antibody retesting at later pregnancy stages. Their median gestational week at retesting was 35 (range 28–41). Twelve of these exhibited IgG antibody seroconversion; five of them had infants with congenital infections, and all were live births. Four were positive for both CMV DNA and viral isolation in urine, whereas one was only positive for CMV DNA. All were asymptomatic (Table 1).

### 3.2. Revised Versus Original Cutoff of CMV IgM Antibody Titer

From September 2013 to March 2017 (the original cutoff period), 1037 and 184 of all 19,435 women had positive/equivocal IgG and IgM above the original and revised cutoff results, respectively. In total, 67 out of the 184 exhibited low avidity, and 8 infants had a congenital infection; 1 of them was symptomatic.

There were no significant differences in the proportions of women with positive/equivocal IgG and IgM above the original or the revised cutoff results among all screened women during the revised and original cutoff periods (Table 3). Moreover, no significant differences in proportions of women with IgM above the revised cutoff (17.6%, the revised cutoff period, and 17.7%, the original cutoff period), low avidity (4.8% and 6.5%), congenital infections including symptomatic infections (1.0% and 0.8%), or symptomatic congenital infections (0.3% and 0.1%) among those with IgM above the original cutoff results (Appendix A). This confirms the validity of the revised cutoff of CMV IgM antibody titer in the maternal serological screening program. The test accuracy of the revised cutoff of the CMV IgM antibody titer for the presence of cCMV in women with low avidity was: 100.0% (15 of 15 cases) in sensitivity, 35.6% (48 of 135 cases) in specificity, 14.7% (15 of 102 cases) in positive predictive value, 100.0% (48 of 48 cases) in negative predictive value, and 42.0% (63 of 150 cases) in accuracy. Required numbers of avidity tests were significantly low (*p* < 0.01 and 0.03, respectively) after the cutoff of CMV IgM antibody titer revision (18.1 and 129.6 avidity tests per one congenital infection case including symptomatic infection, respectively, and 63.5 and 1037.0 avidity tests per one symptomatic congenital infection case, respectively). These confirms the effectiveness of the revised cutoff of CMV IgM antibody titer.

## 4. Discussion

Specific CMV IgM antibody titers become detectable between 0 and 3 weeks, with peak titers observed between 1 and 3 months after primary infection. The presence of CMV IgM antibodies in pregnant women is not specific for primary CMV infections because they may persist for months after a primary infection or occur as a result of assay cross-reactivity. Measurements of IgG antibody avidity as a reflex test are, thus, recommended in women with positive IgG and IgM results to identify non-primary or primary infections. Performance parameters, such as sensitivity, specificity, and positive and negative predictive values, for IgM antibodies for primary infections vary [11]. Recent studies on different commercial CMV IgM assays showed that the concordance of the outcomes for IgM antibodies between those assays is 84–95% [5,12,13,14,15,16,17,18] and that their relative sensitivity and specificity for primary infections are 54–100% and 62–100%, respectively [5,13,19,20,21,22,23,24,25].

CMV IgM antibody tests used in Japan are generally insufficient for the effective diagnosis of primary infections, with the widely used Denka assay also having a low specificity. We previously reported that in cases with low IgG avidity index, the IgM titer tends to be high [6]. CMV IgM titers obtained with different assays in patients with recent primary infections (≤3 months) are higher than those for old infections (>3 months) [14]. Moreover, we found that high Denka IgM titers in women with low avidity are a risk factor for cCMV, as cCMVs occur only in women with titers ≥4.00 [6]. Although we continued our serological study after that report, additional cCMV cases occurred only in women with low avidity and Denka test titers ≥4.00 [9]. Others have reported high Denka assay IgM titers as being a predictor of cCMV despite relatively low sensitivity [7].

The maternal screening program detected congenital infection cases in women with low IgG avidity and high IgM titers (above the revised cutoff in the current study) between September 2013 and March 2017, but no cCMV cases in women with low avidity and low IgM titers (above the original cutoff but under the revised cutoff) [9]. We assumed that this is due to primarily infected women rarely being included in those with low avidity but low IgM titers (under the revised cutoff). We, therefore, revised IgM titer cutoff for IgG avidity measurements as a reflex test after April 2017. During this period, we identified equivalent (not significantly different) numbers of women with low avidity and cCMV.

We have two potential explanations for why women with primary infection are rarely included in those with low avidity and low IgM titers (under the revised cutoff in the current study). One is that some women exhibiting low avidity have been reported to have non-primary infections [5]. Moreover, in non-primary infection cases with very low IgG antibody titers, low avidity can be observed [4]. The other potential explanation is that most women with IgM under the revised cutoff have non-primary infections, and that women who exhibit high avidity are usually considered non-primary infection cases. In our previous cohort, 922 women exhibiting high avidity were diagnosed with non-primary infections [9] and 87.3% had low IgM titers (under the revised cutoff). We, therefore, anticipated almost no primarily infected women among those with low avidity but IgM under the revised cutoff.

A limitation of this study is that we were not able to confirm primary infections serologically. Primary infections are typically confirmed by IgG seroconversion. As low IgG avidity does not always indicate a primary infection, other serological or non-serological tests would have been necessary to show that low avidity and IgM above the original cutoff but under the revised cutoff indicate non-primary infections. The number of neonatal urine CMV DNA tests in women with suspected past infection (i.e., high IgG avidity, positive/equivocal IgG and IgM under the revised cutoff, or positive/equivocal IgG and negative/equivocal IgM) was low. However, the two cCMV infants of women with suspected past infections that we identified both showed high IgG avidity (i.e., high IgG avidity and positive IgG and IgM under the revised cutoff). Therefore, our hypothesis that only high IgM titer women should be tested for IgG avidity is one of the issues that must be explored further.

Neonatal urine CMV DNA screening is more effective than maternal antibody screening in identifying cCMV infants, as maternal antibody screenings cannot detect cCMV infants in mothers with non-primary infection. However, from a different point of view, maternal antibody screenings are able to effectively detect cCMV infants in mothers with primary infection. It is believed that maternal antibody screening can be effective before neonatal urine screening is initiated. We started the maternal antibody screening “CMieV” program in Mie, Japan, in 2013 and have continued it to date. Measurements of IgG avidity were a crucial tool in efficiently identifying cCMV infants in women with primary infection in maternal antibody screening. We would like to emphasize that the present study combines IgG avidity and IgM cutoffs in maternal antibody screening with the main objective of identifying cCMV infants in women with primary infection. This is the first report on the revised cutoff of IgM antibody titer in maternal serological screenings, showing that the revised cutoff was valid and effective in identifying women with primary infections and newborn cCMV infection. We found that it was more effective to identify women with primary infections by administering avidity tests only to women with high CMV IgM titers rather than to all IgM-positive women. Our consistent assertion is that women only with high CMV IgM titers should be tested for IgG avidity in the maternal antibody screening. As the revised CMV IgM titer cutoff improves maternal antibody screening in identifying women with primary CMV infection and newborn cCMV infection, we recommend the continued use or the wider implantation of the revised CMV IgM cutoff for other users of the Denka assay. However, further study related to assays other than the Denka assay is required to apply a revision of CMV IgM titer cutoff to other CMV IgM assays.

## Figures and Tables

**Figure 1 viruses-15-00962-f001:**
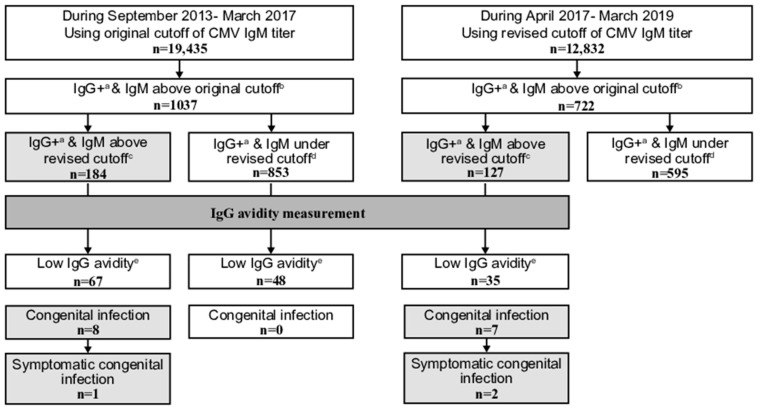
Comparison of positive cytomegalovirus (CMV) immunoglobulin (Ig) G and IgM above the original and revised cutoff results obtained as part of the maternal CMV antibody screening program in pregnant women from September 2013 to March 2017 (using the original cutoff of IgM titer) with those obtained from April 2017 to March 2019 (using the revised cutoff). ^a^ including equivocal IgG. ^b^ ≥1.21 IgM antibody titer. ^c^ ≥4.00 IgM antibody titer. ^d^ <4.00 IgM antibody titer. ^e^ ≤35.0% IgG antibody avidity index.

**Figure 2 viruses-15-00962-f002:**
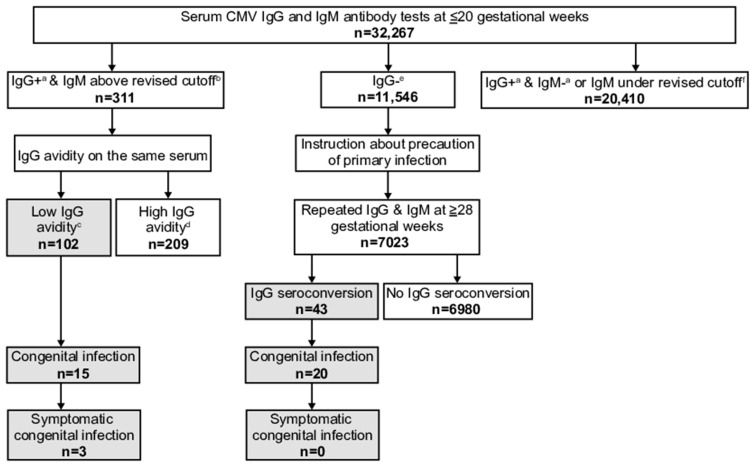
Flowchart of the maternal cytomegalovirus (CMV) antibody screening program from September 2013 to March 2019 (period spanning the original and revised cutoffs), with numbers of screened pregnant women unified to the revised cutoff of CMV immunoglobulin (Ig) M antibody titer. ^a^ including equivocal CMV IgG. ^b^ ≥4.00 CMV IgM antibody titer. ^c^ ≤35.0% CMV IgG antibody avidity index. ^d^ ≥35.1% CMV IgG antibody avidity index. ^e^ with any CMV IgM antibody result. ^f^ <4.00 CMV IgM antibody titer.

**Table 1 viruses-15-00962-t001:** Presentation of the 12 and the other 2 congenital cytomegalovirus (CMV) infection cases from mothers with primary (the revised cutoff period) and non-primary (the original cutoff period) CMV infection, respectively.

No.	Maternal Age (Year)	Maternal Parity (Para)	Result of Maternal Antibody Screening	Titer of IgM at Early Pregnancy (Index)	Avidity Index of IgG at Early Pregnancy (%)	GW of Maternal Antibody Tests in Early/Late Pregnancy (Week)	GW of Delivery (Week)	Delivery Mode	Infant Sex	Infant Body Weight at Birth (g)	Infant Height at Birth (cm)	Infant HC at Birth (cm)	Infant Apgar Scores at 1, 5 min	UA pH	Universal Newborn Hearing Screening	Amount of CMV DNA in Neonatal Urine (log_10_) (Copy/mL)	Viral Isolation of Neonatal Urine	Subunit of gB	Infant Abnormal Neurological Tests
1	24	2	Low IgG avidity	12.00	2.2	10/-	38 + 6	Vaginal	Male	2930	50.0	32.0	9, 10	7.309	Pass	6	+	1	-
2	30	1	Low IgG avidity	8.49	11.4	10/-	39 + 6	Vaginal	Male	3102	50.0	33.4	9, 10	7.349	Pass	4	+	3	-
3	30	0	Low IgG avidity	7.74	31.9	11/-	37 + 5	Vacuum	Male	3090	51.0	33.5	3, 9	7.068	Left refer	4	+	3	MRI + ABR
4	26	0	Low IgG avidity	9.72	0.0	7/-	41 + 3	Vaginal	Male	2930	48.0	32.0	10, 10	7.360	Pass	7	+	3	-
5	22	1	Low IgG avidity	12.00	2.5	13/-	39 + 3	Vaginal	Female	2816	48.3	32.0	9, 10	7.284	Pass	6	+	1	-
6	24	0	Low IgG avidity	12.00	2.5	10/-	40 + 2	Vaginal	Male	3482	49.5	32.0	8, 10	7.236	Bilateral refer	4	-	1	ABR
7	32	1	Low IgG avidity	12.00	4.5	11/-	39 + 4	Vaginal	Female	3008	48.0	33.0	10, 10	7.367	Pass	5	+	1	-
8	20	1	IgG seroconversion	-	-	12/34	39 + 1	Vaginal	Male	3496	50.3	34.0	10, 10	7.340	Pass	6	+	1	-
9	27	1	IgG seroconversion	-	-	11/43 *	38 + 1	Vaginal	Female	2458	49.0	30.0	10, 10	7.460	Right refer	3	+	1	ABR **
10	30	1	IgG seroconversion	-	-	8/36	39 + 5	Vaginal	Male	3512	50.5	33.3	9, 10	7.462	Pass	4	+	3	-
11	22	0	IgG seroconversion	-	-	12/34	41 + 0	Vaginal	Male	4006	51.0	36.0	9, 10	7.276	Pass	6	-	3	-
12	35	1	IgG seroconversion	-	-	16/36	39 + 5	Vaginal	Female	3428	49.0	35.5	8, 9	7.273	Pass	6	+	1	-
13	26	1	High IgG avidity	3.68	76.6	14/-	22 + 1	Abortion	Male	476	25.0	18.0	-	-	-	5 (Ascites)	-(Ascites)	-	-
14	27	2	High IgG avidity	2.17	77.3	9/-	40 + 1	Vaginal	Female	3134	50.0	33.0	10, 10	7.297	Pass	4	+	1	ABR

* IgG seroconversion was confirmed after delivery. ** ABR abnormality was observed due to internal auditory meatus stenosis instead of congenital CMV. GW, gestational week; HC, head circumference; UA, umbilical cord artery; gB, glycoprotein B; ABR, auditory brainstem response; CMV, cytomegalovirus; Ig, immunoglobulin; MRI, magnetic resonance imaging.

**Table 2 viruses-15-00962-t002:** Numbers of pregnant women, neonatal urine cytomegalovirus (CMV) DNA tests, and congenital cytomegalovirus (cCMV) cases according to the results of maternal CMV antibody screening other than low IgG avidity before (the original cutoff period) and after (the revised cutoff period) April 2017.

Results of Maternal CMV Antibody Screening	Number of Pregnant Women	Number of Neonatal Urine CMV DNA Tests	Number of cCMV Cases Including Symptomatic cCMV	Number of Symptomatic cCMV Cases
**Positive * IgG, IgM above the revised cutoff, and high IgG avidity**	209	71	0	0
**Positive * IgG, IgM under the revised cutoff, and high IgG avidity**	805	48	2	2
**Positive * IgG, IgM under the revised cutoff, and unknown IgG avidity**	595	126	0	0
**Positive * IgG and negative * IgM**	18,962	146	0	0

* Including equivocal. Cytomegalovirus, CMV. Immunoglobulin, Ig. Congenital cytomegalovirus, cCMV.

**Table 3 viruses-15-00962-t003:** Proportions of pregnant women with positive (including equivocal) CMV immunoglobulin (Ig) G and IgM above the revised (≥4.00 index) the original cutoff (≥1.21) among all pregnant women during the corresponding periods.

Study Period	April 2017–March 2019	September 2013–March 2017	*p* Value
Cutoff of CMV IgM titer to measure IgG avidity as a reflex test in maternal antibody screening program	The revised cutoff (≥4.00 index) of CMV IgM titer	The original cutoff (≥1.21 index) of CMV IgM titer	
Total number of participating pregnant women	12,832	19,435	
Number of pregnant women with positive CMV IgG * and IgM above the original cutoff (≥1.21 index of IgM titer)	722 (5.6%)	1037 (5.3%)	0.26
Number of pregnant women with positive CMV IgG * and IgM above the revised cutoff (≥4.00 index of IgM titer)	127 (1.0%)	184 (1.0%)	0.73

* Including equivocal CMV IgG. CMV, cytomegalovirus; Ig, immunoglobulin.

## Data Availability

The data presented in this study are available on request from the corresponding author.

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
