# Peer review of "Revision of Cytomegalovirus Immunoglobulin M Antibody Titer Cutoff in a Maternal Antibody Screening Program in Japan: A Cohort Comparison Involving a Total of 32,000 Pregnant Women"

_viruses, 2023, doi:10.3390/v15040962_

Round 1
Reviewer 1 Report
This is a well written prospective study to assess the benefit of a new cut-off value for CMV IgM measurements for further studies to identify primary HCMV infection in expectant mothers to diagnose congenital CMV cases.
I have a couple of errors I have noted related to the PCR methods: The primers and probe for the CMV DNA test are listed but you do not state which protein is used. I am assuming it is gB as you list in table 1 the gB subunits. Following on from this the methods for determing gB subunit is not given.
Reviewer 2 Report
Abstract lines 31 to 33 – the 1% percentages shown in Table 3 should be given here
Abstract line 35 – The word “helps” is a little vague. What is the conclusion? Does the revised cut-ff improve screening? Should it be implemented, or is further work required?
P 2 line 51 – terminology can be improved. Suggest change “Maternal infections before (non-primary) and after (primary infections during pregnancy” to “ Reactivation, reinfection and primary CMV infection during pregnancy…”
P 2 line 75 – To make it clearer to the reader, I suggest including that references 6 and 7 suggested a revised IgM titer cut-off which may be used to reflexly add IgG avidity testing.
P2 line 84 – Antibody titers are usually quantified in units e.g. IU/ml
What units are these numbers referring to? If they are index values, then that needs to be clarified so that the reader is aware the results are kit-specific .
P2 line 86 – why was the cut-off of >4.0 chosen? References 6 and 7 refer to cut-offs of 6.0 and 7.28
P 2 line 90 – is the 35% avidity cut-off from the manufacturer, Enzygnost? Did the manufacturer not give an Indeterminate range? More clarity required here.
P3 line 118 – suggest you change “obstetrical” to “obstetric”
P 3 line 120 – “IgM antibody titer index” doesn’t sound correct – it is either the antibody titer, or and index value.
P 3 line 120 and elsewhere – Suggest change the words “additionally measure” to “reflex test”, if that is what you intended to say.
P4 line 155 – I suggest you add something to the effect of “We retrospectively re-analysed laboratory data using the revised cut-off…” to provide clarity to the reader in the methods section.
P 5 lines 178 and 184 – suggest you add percentages here.
P 9 lines 262 and 271 – I suggest you re-phrase “Primarily infected women” with “ Women with primary infection”
P 9 lines 278 and 280 – I suggest you replace “women without low avidity” with “women with suspected past infection” if that is what you meant.
Supplementary p 2 line 16 – the title needs to be clear, that the figure refers to the time period prior to 2017
OVERALL:
I struggle to see the importance of the research, as the main finding was not clearly emphasized. The performance of the assay was equivalent in a similar population of patients before and after the revised cut-off. Does the researcher then recommend the continued use or the wider implantation of the revised cut-off for other users of the assay? If it is a universal measurement, then should this cut-off be used with assays from other manufacturers too? Or is further work required, if so then what work is required?
